# A Preliminary Study on Quantitative Analysis of Collagen and Apoptosis Related Protein on 1064 nm Laser-Induced Skin Injury

**DOI:** 10.3390/biology13040217

**Published:** 2024-03-27

**Authors:** Qiong Ma, Yingwei Fan, Yufang Cui, Zhenkun Luo, Hongxiang Kang

**Affiliations:** 1Beijing Institute of Radiation Medicine, Beijing 100850, China; maqiong666@163.com (Q.M.); yufangc@vip.sina.com (Y.C.);; 2School of Medical Technology, Beijing Institute of Technology, Beijing 100081, China; fanyingwei@bit.edu.cn

**Keywords:** infrared laser, laser-induced skin wound, collagen, apoptosis-related protein

## Abstract

**Simple Summary:**

With the rapid progress of laser technology and its expanding applications, there is a growing need for detailed information regarding its effects on living tissue and the treatment of skin damage caused by this kind of radiation. Both collagen levels and apoptosis-related protein expressions were quantitatively and comparatively analyzed in porcine skin injuries induced by laser exposure. The findings revealed skin injuries primarily manifested as white spots and burnt spots. There was a progressive increase in the size and depth of skin damage as the laser dosage was elevated. Laser-induced apoptotic cells in porcine skin injuries were significantly increased in the late stages of healing. Subjection to a 1064 nm laser led to a decline in skin healing efficiency as the radiation dosage escalated, leading to disrupted collagen arrangement and inadequate collagen content. Heightened Bax, caspase-3, and caspase-9 protein expressions in healing skin tissues were directly linked to escalated cell apoptosis.

**Abstract:**

To investigate the associated factors concerning collagen and the expression of apoptosis-related proteins in porcine skin injuries induced by laser exposure, live pig skin was irradiated at multiple spots one time, using a grid-array method with a 1064 nm laser at different power outputs. The healing process of the laser-treated areas, alterations in collagen structure, and changes in apoptosis were continuously observed and analyzed from 6 h to 28 days post-irradiation. On the 28th day following exposure, wound contraction and recovery were notably sluggish in the medium-high dose group, displaying more premature and delicate type III collagen within the newly regenerated tissues. The collagen density in these groups was roughly 37–58% of that in the normal group. Between days 14 and 28 after irradiation, there was a substantial rise in apoptotic cell count in the forming epidermis and granulation tissue of the medium-high dose group, in contrast to the normal group. Notably, the expression of proapoptotic proteins Bax, caspase-3, and caspase-9 surged significantly 14 days after irradiation in the medium-high dose group and persisted at elevated levels on the 28th day. During the later stage of wound healing, augmented apoptotic cell population and insufficient collagen generation in the newly generated skin tissue of the medium-high dose group were closely associated with delayed wound recovery.

## 1. Introduction

The 1064 nm laser possesses distinct attributes such as a lengthy wavelength, deep-reaching capability, and the capacity to be absorbed by melanin, oxyhemoglobin, and water within skin tissue. The 1064 nm laser is able to enter the tissue more deeply and has greater ablative capacity. The laser is similarly able to penetrate deeper into the skin when acting on skin tissue [1,2,3]. The study of the interaction mechanism between 1064 nm lasers and skin tissue is a very complex and wide-ranging topic in research areas [4,5,6,7,8,9] such as laser surgery, laser thermotherapy and laser imaging, which has become a difficult problem at the intersection of laser physics and biomedicine. This laser finds widespread employment in skin ailment therapy, with its impact on skin integrity forming a pivotal facet of laser-based biological investigations. As early as 1965, Polanyi et al. [10] used a high-power continuous 1064 nm laser to irradiate human tissues, causing vaporization and evaporation of the irradiated tissues to achieve the effect of cutting the tissues and, at the same time, using a low-power laser to irradiate the cutting site to coagulate the tissues in order to achieve the effect of hemostasis. Dang et al. [11] established that non-ablative treatment utilizing a 1064 nm laser can heighten skin suppleness and amplify collagen content within murine skin tissue, indicating that the laser’s photothermal effect fosters the production of type I collagen. Employing a 1064 nm laser, Kim et al. [12] eliminated the epidermis and dermis of rats by means of photothermal decomposition to ablate benign skin anomalies. Their perspective posits this laser as a viable choice for managing benign skin disorders.

Given the rapid development of laser technology and the broad spectrum of laser applications in the biomedical, industrial, military, and other fields [13,14,15,16], inadvertent laser-induced harm is a recurring occurrence [17,18], with skin impairment trailing only ocular injuries. The 1064 nm infrared laser is not visible to the naked eye, thus leading to easier accidental injury. Imprecise control of laser dosage during therapy can result in minor skin damage [19,20]. Laser thermal damage may destroy basal cells and other related tissues that are closely associated with regeneration, resulting in delayed healing or even non-healing of the wound as well as severe scarring at a later stage [21]. Concurrently, utilization of high-powered lasers has the potential to inflict skin harm, especially when safety measures during laser production and development are lax or when operators lack sufficient awareness of protective protocols, leading to adverse consequences. Laser radiation can cause serious damage to human skin tissue such as ablation, carbonization, vaporization, coagulation, and sputtering [22]. The influence of elevated infrared doses is fundamentally linked to robust temperature escalation, and the extent of skin damage correlates closely with infrared dose magnitude [23]. Rationally and effectively harnessing the thermal energy generated by the interplay between lasers and biological tissue to attain the sought-after thermal impact or damage extent is pivotal for laser-based diagnosis and disease management. However, forestalling inadvertent harm to neighboring normal tissues arising from excessive heat stands as the most pivotal facet of laser medicine necessitating safeguarding. Notably, the literature still presents a scarcity of investigations concerning the effects of laser-induced skin damage and the attendant repair traits. Hence, delving into the precise molecular mechanisms underpinning skin injury and repair assumes utmost significance.

In general, skin wound repair is a complex biological process governed by a multitude of factors [24,25], encompassing the release of diverse cytokines, chemokines, and inflammatory agents. Simultaneously, it relies on the synergy between distinct cells to cohesively regulate wound recuperation. Investigations reveal that growth factors, encompassing fibroblast growth factor (FGF) and epidermal growth factor (EGF), within tissues during acute skin injuries sustain tissues’ functional efficacy and participate in the healing trajectory [26]. Other inquiries have underscored the role of the PI3K/Akt signaling pathway in governing cell proliferation, differentiation, and apoptosis and glucose metabolism. Activation of this pathway expeditiously advances wound healing by enhancing cell proliferation and invasion [27]. Notably, certain analyses assert the pivotal regulatory function of fibroblast proliferation, migration, activation, and apoptosis in skin wound repair, particularly underscoring the significance of fibroblast apoptosis during the later phases of healing [28]. Conversely, augmented fibroblast apoptosis during the later stages of injury repair contributes to the restoration of skin integrity [29]. While collagen synthesis and extracellular matrix accumulation represent pivotal facets of wound healing, excessive deposition can precipitate the development of skin scars [30,31,32]. Currently, there is little research on the biological processes involved in the repair of laser-induced skin damage.

Investigations have highlighted the intrinsic link between the healing course of skin injuries and apoptosis, which is modulated by intracellular apoptotic regulatory proteins. The bax and caspase families play a critical role in mitochondria-mediated apoptosis [33,34,35]. Our prior research efforts have elucidated certain characteristics of skin injury and repair resultant from laser exposure [36,37]. The study of the quantity–effect relationship between lasers and skin tissues is an important part of the analysis of the mechanism of the biological action of lasers and also is the basis for clinical application. To delve deeper into the mechanisms governing laser-induced skin injury and subsequent repair, this study examined changes in collagen architecture and the expression of apoptosis-associated proteins, namely Bax, caspase-3, and caspase-9, within porcine skin wounds during the ensuing wound healing post-laser irradiation. This study provides experimental ideas and methods for future exploration of the consequences and mechanisms of cutaneous thermal injuries triggered by a 1064 nm infrared laser. 

## 2. Materials and Methods

### 2.1. Experimental Animals

Five Guizhou miniature fragrant pigs of both sexes (3 females and 2 males), characterized by white hair and weighing 20~25 kg each, were procured from Beijing Liulihe Kexing Experimental Animal Breeding Center (Animal production license No. SCXK (Beijing, China) 2017-0003). These pigs were housed at the Beijing Institute of Radiation Medicine Experiment Animal Center. A standard feeding regimen was followed for 3 to 7 days before the experiment, with the stipulation that only pigs displaying no abnormalities could participate in the study. Preceding and following the experiment, individual pig accommodations were provided. Ethical clearance was obtained from the Beijing Institute of Radiation Medicine Experiment Animal Center in accordance with approved animal protocols (IACUC-DWZX-2019-502).

### 2.2. Laser Radiation Method

Prior to laser exposure, the miniature pigs were anesthetized using intramuscular injection of a 3% sodium pentobarbital solution at a dosage of 1 mL/kg. Subsequently, the intended injury site was shaved. A laser irradiation region measuring approximately 15 cm by 30 cm was marked out on the right side of the pig’s spine, with a grid pattern of 5 × 10 arrays established. A 1064 nm continuous laser was emitted through fiber conduction from a self-developed laser system. The laser output power ranged from 12 W to 955 W and was notably stable, exhibiting a robust linear correlation with the current (linear correlation coefficient *R*^2^ = 0.9999). The laser exposure time (t) was set to 0.5 s. Laser power (P) was gauged using a laser power meter (BGS6333-P400, Beijing Monochrome Optoelectronic Technology Co., Ltd., Beijing, China), and the irradiation spot’s diameter (d) was standardized at 1 cm. The laser irradiation power density (H) was computed using the parameters above, utilizing the equation $H=\frac{4Pt}{\pi d^{2}}$. In the 5 × 10 grid arrays, each grid location represented an irradiation point, with identical laser dosage administered to the five grids within a column. The dosage spectrum ranged from 29.4 J/cm^2^ to 708.2 J/cm^2^, with laser power sequentially escalating from low to high. Prior investigations had indicated that doses below 126.4 J/cm^2^ caused primarily reversible erythema reactions that vanished within seconds or minutes [36]. Hence, for this study, low, medium, and high doses (126.4 J/cm^2^, 320.3 J/cm^2^, and 514.3 J/cm^2^) were selected based on the severity of the injury. A locating ring was utilized to ensure a consistent distance of 10 cm between the laser device tip and the skin surface. The optical path of laser-induced skin damage in miniature pigs is depicted in Figure 1.

### 2.3. Method to Observe Damage Spots

Following 1064 nm laser irradiation at 6 h and 3, 7, 14, and 28 days, one pig was anesthetized for reexamination. The healing regions of skin wounds were macroscopically examined and photographed for measurement. Exudation, bleeding, crusting, or debridement of the skin wound surface was recorded to provide a reference for histopathological analysis. To avert potential harm to animals during handling and immobilization, which could lead to wound tearing, and considering the elevated risk of animal mortality from multiple anesthesia sessions, the experiment adopted a staggered approach for observing different animals at distinct time points. This prevented continuous observation of the same animal over the full 28-day span. One pig was humanely euthanized to obtain samples at a particular time point, yielding five samples at identical irradiation doses. The wound area was sampled along with the surrounding normal tissue. The normal skin of each animal at the unirradiated site was used as a control in this study. These wound samples were fixed in 4% neutral formaldehyde solution; these pre-treated samples were sent to Wuhan Servicebio Biotechnology Company. The company provides a standardized procedure to complete tissue paraffin embedding, sectioning, hematoxylin–eosin (HE) staining, and later Tunel staining and immunohistochemical staining. The resulting pathological sections were examined under a digital pathology system (Pannoramic MIDI, 3DHISTECH Int.) to assess the morphological changes in cutaneous wounds.

### 2.4. Quantitative Analysis of Collagen in Skin Wounds

Paraffin sections of wound samples underwent Sirius Red staining. Using a polarized light microscope, the structure and appearance of type I and type III collagen were analyzed. Type I collagen appeared as coarse, mature, red or orange fibers, while type III collagen manifested as slender, immature, green fibers. After 1064 nm laser irradiation at 6 h and 3, 7, 14, and 28 days, five images with equal fields of view (FOVs) were randomly captured from each sample under consistent sampling conditions. Comparable quantitative collagen analysis was performed on normal skin tissues extracted from each pig’s back. The collagen’s area density percentage was computed for the stained images using the following formula: positive area of type I collagen + positive area of type III collagen)/total image area × 100%. These areas were manually selected and quantified using ImagePro Plus 6.0 (as depicted in Figure 2), and the software was employed to calculate the average value.

### 2.5. Apoptosis Detection by TUNEL Staining

Paraffin sections were prepared from the affected tissue spots for TUNEL staining. This technique, an in situ terminal transferase labeling method, accurately portrays the primary biochemical and morphological attributes of cellular apoptosis by staining apoptotic cell nuclei or apoptotic bodies. Components including protease K (G1234), TUNEL kit (G1507), DAB chromogenic agent (G1212), and hematoxylin staining solution (G1076) were acquired from Wuhan Servicebio Biotechnology Co., Ltd., Wuhan, China. At 6 h and 3, 7, 14, and 28 days post-1064 nm laser irradiation, apoptosis occurrences were visualized within the wound tissue under a light microscope. Apoptotic cells exhibited positive brown staining, with blue nuclei.

### 2.6. Immunohistochemical Detection

Subsequent to paraffin section preparation from damaged tissue spots, the expression levels of Bax, caspase-3, and caspase-9 were assessed through immunohistochemical (IHC) staining. Levels of caspase-9, which functions downstream of mitochondria within the intrinsic pathway, and caspase-3, which serves as an effector caspase during the final stages of apoptosis, were evaluated. Additionally, Bax, a proapoptotic member of the Bcl-2 family pivotal in intrinsic apoptotic signaling regulation, was examined. Rabbit anti-mouse polyclonal antibodies for Bax (bs-0127R), caspase-3 (bs-0081R), and caspase-9 (bs-20773R) were sourced from Beijing Boaosen Biotechnology Co., Ltd., Beijing, China. Within wound tissue, brownish-yellow granules in the cytoplasm or nucleus were examined under a light microscope. At 3, 7, 14, and 28 days subsequent to 1064 nm laser irradiation, five images (magnification of 400) with consistent fields of view (FOVs) were randomly captured from the wound region under identical sampling conditions. These images were analyzed using Image-Pro Plus 6.0, yielding their integrated optical density (IOD) of positive protein expression, as illustrated in Figure 2. Then the mean and standard deviation of IOD were calculated using GraphPad Prism 9.0 software. Based on the IOD value of the normal group, the relative positive protein expression value was determined by calculating the ratio of IOD in the irradiation group to that in the normal group. Subsequently, the average value was computed, and statistical analysis was performed using the software.

### 2.7. Statistical Analysis

The results were subjected to analysis using GraphPad Prism 9.0, presented as $\bar{x}\pm s$. Group comparisons were conducted through two-way ANOVA (Tukey’s test). Statistical significance was set at *p* < 0.05.

## 3. Results

### 3.1. Macroscopic Observation of the Wound Healing

Upon exposure to varying doses of the 1064 nm laser for 0.5 s, macroscopic observation of pig skin injury and healing was documented, as depicted in Figure 3. The skin injuries primarily manifested as white spots and burnt spots, with disparities evident among different animals even at the same dose. An assessment 1 h post-laser exposure unveiled minor white spot reactions in the group receiving the 126.4 J/cm^2^ dose, while the 320.3 J/cm^2^ dose group exhibited predominantly burnt spot reactions, interspersed with some white spots. The group subjected to the 514.3 J/cm^2^ dose exhibited mainly burnt spot reactions. Evidently, escalating doses correlated with a noteworthy increase in injury area. Evaluation of laser-induced skin injuries 3 to 28 days post-irradiation indicated the emergence of thin scabs around some wounds after 3 days. Nevertheless, distinct healing patterns were not conspicuous among radiation groups. On days 7 and 14 post-irradiation, discernible degrees of wound healing were evident across all irradiation groups. While the low-dose group exhibited evident wound surface contraction, healing momentum markedly slowed down in the medium- and high-dose groups. By day 28, significant recovery was apparent in the low-dose group. In contrast, the medium- and high-dose groups exhibited incomplete recovery, characterized by gradual wound shrinkage and healing.

Morphological examination of pig skin injury and healing at 6 h and 28 days post-irradiation is presented in Figure 4. At the 6-h mark after irradiation, skin injury displayed a gradual increment in correlation with augmented laser irradiation doses, particularly notable in the depth of injury. Notably, in the group subjected to the 320.3 J/cm^2^ dose, the skin wound site exhibited separation between the epidermis and dermis. In the group exposed to the 514.3 J/cm^2^ dose, the epidermis and a section of the dermis were absent due to laser-induced vaporization. At the 28-day point post-irradiation, scabs on the wound surface remained adhered, with a corresponding decline in skin healing as irradiation dose escalated. Although the wound surface in the 126.4 J/cm^2^ and 320.3 J/cm^2^ dose groups was entirely enveloped by the newly formed epidermis and dermis, and displayed a thicker layer. Conversely, the group receiving the 514.3 J/cm^2^ dose showed incomplete wound filling, with the surface inadequately covered. The third-row image in Figure 4 reveals pronounced late-stage granulation tissue underneath the newly formed epidermis in the medium-high dose groups, featuring abundant spindle-shaped fibroblasts. Normal tissue exhibited robust, densely arranged collagen fibers, while the newly formed collagen fibers in dermal tissue from the low-, medium-, and high-dose groups appeared slender and loosely arranged. 

### 3.2. Structural and Quantitative Analysis of Collagen in the Skin Wounds after Irradiation

Figure 5 illustrates the structural analysis of collagen (type-I and type-III collagen) in pig wounds at different time points after laser irradiation. The focus of the analysis lies on the newly formed tissue regions within the wounds. Normal skin tissue showcased intact collagen structures, characterized by closely arranged red, thick fascicular type I collagen fibers that formed a grid, interspersed with green, slender type-III collagen fibers. At 6 hours post-irradiation, collagen structures in all irradiation groups were subject to severe damage, resulting in a sharp decline in collagen content with increasing irradiation doses. At 3 days post-irradiation, collagen structures across all irradiation groups displayed looseness and disorder, accompanied by a significant reduction in collagen content. This reduction was particularly pronounced in the medium-high dose groups (320.3 J/cm^2^ and 514.3 J/cm^2^), as compared to the low-dose group (126.4 J/cm^2^). Collagen structures in wound areas remained severely compromised at 7 days post-irradiation, characterized by a scarcity of new fibers. By day 14, collagen production had increased across all dose groups, particularly evident as robust type I collagen in the low-dose group, whereas the medium- and high-dose groups predominantly displayed type III collagen structures. However, collagen content in all dose groups remained below that observed in the normal group. By day 28, collagen structures had significantly recovered across all dose groups, accompanied by a notable increase in collagen content. Nevertheless, the medium-high dose groups still harbored numerous slender, immature, green type III collagen fibers. Despite the improvements, collagen structure and content in all dose groups had not fully normalized to the levels observed in the normal group. These findings indicate that newly formed collagen fibers extended from undamaged tissue towards the wound center. Consequently, collagen content at the wound center was notably lower, particularly evident in the high-dose group at both 14 and 28 days post-irradiation.

Furthermore, the quantification of collagen in the wounds is presented in Figure 6. The area density percentage of collagen in normal skin was approximately (13.1 ± 1.3)%. Here, the area density content signifies collagen concentration within the skin tissue. At 6 hours post-irradiation, collagen content in the 126.4 J/cm^2^ group approximated 95% of that of the normal group (12.4% vs. 13.1%, *p* = 0.07). However, due to worsening skin injury, collagen content in other dose groups notably decreased in comparison to the normal group (*p* < 0.01), and the 514.3 J/cm^2^ dose group displayed the worst collagen content in comparison to other dose groups (*p* < 0.01) with only (2.8 ± 0.9)% collagen content. By 3 days post-irradiation, collagen content in each dose group reached its lowest point, even the low-dose group’s collagen content being merely 46% of the normal group (6.0% vs. 13.1%, *p* < 0.01). Collagen content in the medium- and high-dose groups notably decreased in comparison to the low-dose group (*p* < 0.01). At 7 and 14 days post-irradiation, collagen content gradually increased in each irradiation group, with the low-dose group’s collagen content reaching (11.4 ± 1.7)%, significantly different from the normal group (*p* < 0.01). However, the collagen content of the medium- and high-dose groups remained significantly lower than that of the normal or low-dose group (*p* < 0.01). By day 28, collagen content had continued to rise in all irradiation groups, with the area density percentage of collagen in the low-, medium-, and high-dose groups measuring (11.9 ± 1.7)% (91% of the normal gruop), (7.6 ± 2.0)% (58% of the normal group), and (4.8 ± 1.3)% (37% of the normal group), respectively. In particular, collagen content in the high-dose group was significantly lowest in comparison to the normal and low- and medium-dose groups (*p* < 0.01). These findings underscore that, even 28 days post-irradiation, the collagen content in the wound tissues of each irradiation group had yet to be restored to normal tissue levels.

### 3.3. Apoptosis Analysis of Radiation Lesions

Post-0.5-s 1064 nm laser irradiation, the outcomes of cell apoptosis in skin wounds are displayed in Figure 7. Low, medium, and high doses of laser irradiation induced necrotic damage to skin tissues, consequently leading to minimal new tissue and apoptotic cells in skin wounds at 3 days post-irradiation. At 7 days post-irradiation, apoptotic cells emerged within the newly formed epidermis of the 126.4 J/cm^2^ dose group. In contrast, severe damage and limited repair in the 320.3 J/cm^2^ and 514.3 J/cm^2^ dose groups resulted in insignificant apoptotic cell presence in wound tissues. By 14 days, the medium- and high-dose groups exhibited significant increases in apoptotic cells within the new epidermis and late subepidermal granulation tissue, primarily involving fibroblasts cells. At 28 days post-irradiation, the low-dose group still had many apoptotic cells in the newly formed epidermal tissue, and the 320.3 J/cm^2^ and 514.3 J/cm^2^ groups maintained many apoptotic cells within the newly formed dermal tissue. These results underscore the necessity of fibroblast and vascular endothelial cell apoptosis in granulation tissue for effective skin wound healing, while excessive apoptosis proves detrimental to the healing process.

### 3.4. Expression of Bax Protein in Skin Wounds after Irradiation

Following 0.5-s 1064 nm laser irradiation, the results pertaining to Bax protein expression in skin wounds are demonstrated in Figure 8. The relative value of Bax protein’s integrated optical density (IOD) in the normal group measured (1.00 ± 0.209). By 3 days post-irradiation, due to severe tissue degeneration and necrosis within skin wounds across all irradiation groups, Bax protein expression experienced a transient decrease in the 320.3 J/cm^2^ and 514.3 J/cm^2^ dose groups (*p* < 0.01). At 7 days post-irradiation, Bax protein expression in newly formed skin tissue proved slightly elevated in the 126.4 J/cm^2^ dose group compared to the normal group; however, statistical significance was lacking. Conversely, insufficient repair of skin wound damage led to significantly lower Bax protein expression in newly formed skin tissue in the 320.3 J/cm^2^ and 514.3 J/cm^2^ dose groups compared to the normal and low-dose groups (*p* < 0.01). At 14 days, apparent wound repairs were observed across all irradiation groups. Bax protein expression in newly formed skin tissue decreased and neared normal levels in the low-dose group. In contrast, its expression significantly increased in the medium- and high-dose groups, notably surpassing the normal group (*p* < 0.01). Concurrently, Bax protein expression in the 514.3 J/cm^2^ dose group was significantly highest in comparison to the normal, low- and medium-dose groups (*p* < 0.01). By 28 days, Bax protein expression in the 514.3 J/cm^2^ dose group remained significantly higher than in the normal and low- and medium-dose groups (*p* < 0.01). These findings underline the alignment between the trend of Bax protein expression changes within newly formed skin tissue and the aforementioned variations in cell apoptosis.

### 3.5. Expression of Caspase Proteins in Skin Wounds after Irradiation 

The outcomes related to caspase-9 and caspase-3 protein expression in skin wounds from exposure to 0.5-s 1064 nm laser irradiation are portrayed in Figure 9. Within the normal group, caspase-9 and caspase-3 protein expressions were standardized, with relative IOD values of 1.00 ± 0.179 and 1.00 ± 0.325, respectively. By the third day post-irradiation, transient declines in caspase-9 protein expression were evident across all irradiation groups (*p* < 0.01). Concurrently, caspase-3 protein expression in the 320.3 J/cm^2^ and 514.3 J/cm^2^ groups decreased in comparison to the normal and low-dose groups; however, statistical significance was lacking. This can be attributed to early-stage severe tissue degeneration and necrosis within skin wounds across all irradiation groups. By 7 days post-irradiation, caspase-9 and caspase-3 protein expression gradually ascended across all irradiation groups. In the 126.4 J/cm^2^ dose group, caspase-9 and caspase-3 expression in newly formed skin tissue surpassed that of the normal group; however, statistical significance was lacking. Conversely, due to incomplete damage repair, caspase-9 protein expression in the 320.3 J/cm^2^ and 514.3 J/cm^2^ groups, and caspase-3 protein expression in the 514.3 J/cm^2^ group, notably lagged behind the normal group (*p* < 0.01) and even the low-dose group (*p* < 0.01). By 14 days, the low-dose group experienced negligible alterations in caspase-9 protein expression, remaining similar to the 7-day level. Nevertheless, while caspase-3 protein expression declined slightly from the 7-day level, it remained higher than the normal group. Meanwhile, medium- and high-dose groups showed significant escalations in caspase-9 protein expression in newly formed skin tissue, surpassing the normal group (*p* < 0.01). Concurrently, caspase-9 and caspase-3 protein expression in the 514.3 J/cm^2^ group notably increased in comparison to the normal, low- and medium-dose groups (*p* < 0.01). At 28 days post-irradiation, caspase-9 protein expression in the low-dose group approximated normal levels. High-dose group caspase-9 expression and medium- and high-dose group caspase-3 expression decreased compared to the 14-day interval, yet still notably surpassed normal group levels (*p* < 0.01) and even the low-dose group (*p* < 0.01 or *p* < 0.05). In particular, caspase-9 and caspase-3 protein expression in the high-dose group was significantly highest in comparison to the normal, low- and medium-dose groups (*p* < 0.01). The findings underscore the substantial alignment between trends in Bax, caspase-3, and caspase-9 protein expression within newly formed skin tissue.

## 4. Discussion

For a considerable duration, the focus of laser injury has predominantly been on ocular injury. Nevertheless, there exist limited literary reports on the process of laser-induced skin injury and subsequent healing in animals. Being the outermost organ of the human body, the skin is more prone to thermal damage when compared to other tissues. When subjected to laser hyperthermia, the skin’s surface temperature experiences an instantaneous elevation to a specific level, consequently predisposing it to skin burns and the initiation of skin collagen denaturation. Alterations in the configuration of proteins and organelles within biological tissues can give rise to cell demise or even tissue necrosis. In this investigation, miniature pigs, exhibiting skin structures akin to those of humans, were employed as the experimental subjects, owing to their resemblant skin physiological structures, epidermal morphology, renewal cycles, and wound mending patterns. The epidermal composition of miniature pigs encompasses the basal layer, spinous layer, granular layer, transparent layer, and stratum corneum, while the dermal layer comprises the papillary and reticular layers, coupled with subcutaneous tissue and skin appendages. Moreover, the body fluids and metabolic processes subsequent to skin burns manifest striking similarities. Thus, miniature pigs present a more exemplary animal model for investigating laser burns [38,39]. In this experiment, the light source employed was a 1064 nm laser, which illuminated the experimental pigs at varying doses. The alterations in collagen structure and apoptotic cells within skin wounds were meticulously observed alongside the quantitative analysis of collagen content and the expression of apoptosis-related proteins within skin lesions.

Skin injury and wound healing represent intricate biological processes, typically encompassing the hemostasis, inflammation, tissue regeneration, remodeling, and maturation stages [40,41]. Earlier investigations have revealed a lack of discernible hemostasis or exudation phases in the recuperative trajectory of mouse skin trauma induced by lasers. This distinguishes it from conventional skin trauma or burns, constituting a crucial distinction [36]. Some studies have similarly found differences in skin damage and wound healing between laser irradiation and thermal burns. Zhang et al. [42] similarly observed some difference between laser irradiation and thermal burns in terms of skin damage and wound healing. Findings from this study illustrated the dry nature of pig skin wound surfaces, with no signs of inflammatory exudates or bleeding 3 days following laser exposure. The wound healing journey of pig skin injuries also lacked conspicuous hemostasis or exudation periods. Laser-triggered skin injuries predominantly manifested as instances of acute coagulation necrosis, characterized by white spots and scorched areas. Remarkably, throughout the healing course, all wounds remained uninfected and non-suppurative, potentially differing from responses seen in skin trauma or burns. Within this study, the size and depth of skin injuries following laser treatment exhibited a positive correlation with the administered irradiation dose. Specifically, injuries within the low-dose group were primarily confined to the epidermal and superficial dermal layers, while the medium- and high-dose groups displayed injuries that extended to the subcutaneous tissue layer. After a period of 28 days post-irradiation, the white spot injuries induced by low-dose lasers exhibited near-complete macroscopic recovery, closely resembling the normal group. Conversely, burn injuries inflicted by medium- and high-dose lasers still prominently featured scabs on the wound surfaces. The recuperation of skin wounds beneath these scabs lagged behind that of the normal group. Notably, even the skin surface of the high-dose group had not achieved complete closure.

Collagen creation following skin trauma emerged as a pivotal factor influencing the recuperation of skin injuries. The skin’s collagen primarily comprises type I and type III collagen. Type I collagen contributes to the structural integrity of the skin [43]. Type III collagen production surges in the initial phases of wound healing, significantly contributing to collagen reconstruction and repair. In the early stages post-laser irradiation (3 days), collagen within each irradiation group exhibited degradation and necrosis, inducing substantial impairment to the collagen’s structural integrity, consequently leading to a marked decline in collagen content. In the intermediate stage post-laser irradiation (7–14 days), collagen structure began to regenerate, characterized mainly by slender type III collagen fibers, and collagen content exhibited gradual augmentation. In the later stage post-laser irradiation (28 days), coarse type I collagen escalated across each irradiation group, yielding an uneven and lax collagen structure. Despite the increase, collagen content remained significantly inferior to that of the normal group. Within the recovery process of laser-induced skin injuries by means of medium- and high-dose exposures, the synthesis of skin tissue collagen encountered pronounced inhibition, and this was worse in the high-dose group. Consequently, the collagen structure and content within wounds resulting from these exposures struggled to fully revert to normal levels. Hence, it is postulated that the scarcity of collagen during the intermediate and later stages of laser irradiation constitutes a pivotal determinant behind the sluggish wound healing observed within medium- and high-dose groups. Furthermore, select investigations have discovered elevated collagen content in skin burns or trauma compared to normal skin tissue during the late phases of recovery, particularly during scar formation. However, this experimentation revealed lower collagen content in laser-induced skin injuries than the standard levels. This disparity could stem from the comparatively tardy recovery of laser-induced skin injuries in contrast to non-laser-induced skin burns or trauma. Specifically, even 28 days after laser irradiation, noticeable late-stage granulation tissue persisted within the skin wounds of medium- and high-dose groups, marked by the vigorous proliferation of fibroblasts and inadequate collagen fiber synthesis. This serves as an indication that wound healing had yet to fully reach the maturation and remodeling stages. The outcomes stated above necessitate further research and validation. The conjecture remains that insufficient collagen synthesis stands as an important factor contributing to delayed wound healing following medium- and high-dose laser irradiation.

The progression of wound healing is influenced by numerous factors, including radiation, which can disrupt the normal function of fibroblasts within skin tissue. Earlier explorations into the mechanics of wound healing in the context of radiation-induced wounds unveiled the involvement of apoptosis across the entirety of the wound healing process. Notably, the distinctive features included an earlier onset and a prolonged subsistence of apoptosis. Moreover, excessive apoptosis was deemed a primary contributor to delayed wound healing [44]. Further investigations highlighted the crucial roles of the proapoptotic protein Bax and the anti-apoptotic protein bcl-2 in regulating apoptosis within skin wounds and the subsequent healing processes [45]. Presently, the multifaceted regulatory mechanism of the apoptosis pathway is still under study. However, the caspase-dependent pathway has been distinctly recognized, with the bcl-2 and caspase families emerging as two pivotal protein families deeply involved in this process. Functioning as a cluster of cysteine proteases with a specific affinity for aspartic acid cleavage, the caspase family constitutes a protease system that directly precipitates the disintegration of apoptotic cells, thereby assuming a central role in transmitting apoptotic signals. The apoptotic pathways are primarily classified into endogenous and exogenous pathways. Within this framework, caspase-9 assumes the role of an essential catalyst within the endogenous apoptotic pathway. Once activated, it proceeds to engage downstream effectors, such as caspase-3, which in turn targets key cellular structures and regulatory proteins to facilitate apoptotic cell demise [45,46]. In contrast, the exogenous apoptosis pathway hinges upon death receptors located on the cellular membrane. Activation of this pathway necessitates the involvement of caspase-8, culminating in the activation of caspase-3 and consequent initiation of cellular apoptosis [47].

However, there has been a lack of conclusive reports on cellular apoptosis and its associated regulatory proteins within the course of laser-induced skin injuries and subsequent repair. This experiment uniquely examined alterations in cellular apoptosis throughout the process of laser-induced skin injuries and healing in miniature pigs, providing preliminary insights into the functions of select apoptosis-related proteins. The findings indicated that in the initial stage (3 days) following laser exposure, the skin wound tissue exhibited minimal cell apoptosis due to acute tissue necrosis. Transitioning to the intermediate stage (7 and 14 days) after irradiation, apoptotic cells surfaced within the newly forming epidermis of the skin wounds, exemplified by occurrences in the low-dose group of 126.4 J/cm^2^ on the 7th day. Simultaneously, a significant rise in apoptotic cells was observed within the emerging dermis of the skin wounds, particularly involving fibroblast cells. This phenomenon was evident in the medium-dose group of 320.3 J/cm^2^ and the high-dose group of 514.3 J/cm^2^ on the 14th day. As the irradiation progressed to the late stage (28 days), corresponding with the phase of skin wound remodeling, the new epithelial layer largely enveloped the wound, leading to a decline in apoptotic cell count. Although the apoptosis level in the low-dose group closely resembled that of the normal group, the medium- and high-dose groups maintained significantly elevated apoptosis rates compared to the normal group. During the phases of skin wound healing, the expression of proapoptotic proteins—namely, Bax, caspase-3, and caspase-9—experienced substantial upregulation, with synchronous alterations among the three proteins. Hence, it is hypothesized that at distinct stages of wound healing, when cell numbers within newly forming skin tissue necessitate augmentation or reduction, apoptotic signaling molecules stimulate the expression of Bax, caspase-3, and caspase-9 proteins through both intracellular and extracellular signal transduction pathways. Caspase-3 regulation is influenced by both endogenous and exogenous apoptotic signaling molecules. Consequently, this experimentation discerned more pronounced shifts in caspase-3 expression, suggesting its potential role in directly facilitating cellular apoptosis. During the intermediate and later phases (14 and 28 days) of wound healing, the heightened expression of Bax, caspase-3, and caspase-9 proteins in the nascent tissues of the medium- and high-dose groups culminated in excessive cellular apoptosis, potentially hindering the progress of skin wound healing. Thus, the conjecture emerges that excessive cellular apoptosis constitutes an additional important factor contributing to the delayed wound healing observed in the wake of medium- and high-dose laser irradiation.

## 5. Conclusions

In conclusion, the recovery process from burn injuries caused by medium- and high-dose lasers exhibited sluggishness. Following 1064 nm laser irradiation during the skin wound healing journey, the collagen content within the skin wounds was found to be insufficient, and its structural assembly was suboptimal. The heightened presence of proapoptotic proteins played a role in promoting apoptosis among fibroblasts and vascular endothelial cells within granulation tissue. It is postulated that apoptosis could impact the trajectory and prognosis of skin wound healing, with both inadequate collagen production and excessive apoptosis contributing to delayed or challenging recuperation in the context of laser-induced skin wounds. Notably, the healing progression and condition of laser-induced skin injuries were notably superior in the low-dose group compared to the medium- and high-dose groups. In particular, the mitigation of excessive apoptosis induced by high doses assumes paramount significance in the realm of drug research and development geared towards the prevention and treatment of laser-induced skin injuries.

## Figures and Tables

**Figure 1 biology-13-00217-f001:**
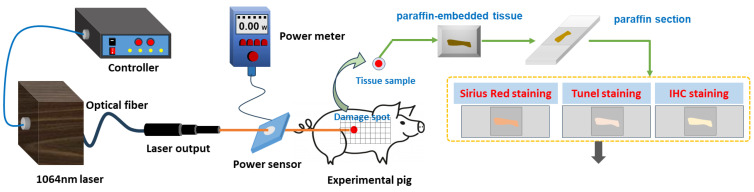
Laser radiation and detection methods of laser-induced skin damage in miniature pigs.

**Figure 2 biology-13-00217-f002:**
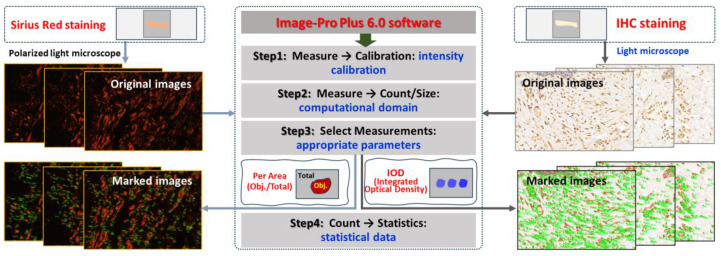
Software-driven quantitative analysis of collagen and apoptosis-related protein in skin wounds.

**Figure 3 biology-13-00217-f003:**
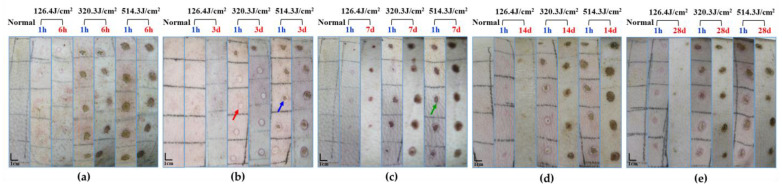
Macroscopic depiction of cutaneous wound healing in miniature pigs exposed to a 1064 nm laser. (**a**) Experiment pig 1, observed 1 h and 6 h post-irradiation prior to euthanasia. (**b**) Experiment pig 2, observed 1 h and 3 days post-irradiation prior to euthanasia. (**c**) Experiment pig 3, observed 1 h and 7 days post-irradiation prior to euthanasia. (**d**) Experiment pig 4, observed 1 h and 14 days post-irradiation prior to euthanasia. (**e**) Experiment pig 5, observed 1 h and 28 days post-irradiation prior to euthanasia. Red arrows denote white spots, blue arrows denote small burnt spots at the center of white spots, and green arrows denote large burnt spots.

**Figure 4 biology-13-00217-f004:**
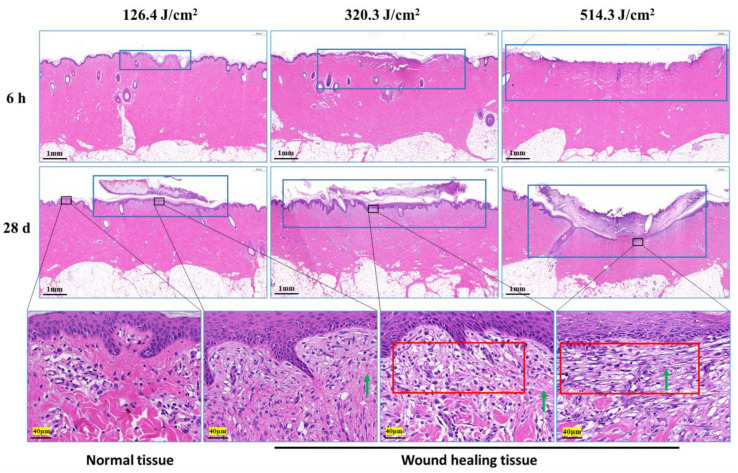
Pathological observation of skin wound healing in miniature pigs exposed to a 1064 nm laser. The images in the third row correspond to magnified views marked by boxes in the second row, and the magnification is 25. Blue boxes denote skin wound sites. Red boxes denote late-stage granulation tissue. Green arrows indicate newly synthesized collagen fibers (red, long or filamentous fibers).

**Figure 5 biology-13-00217-f005:**
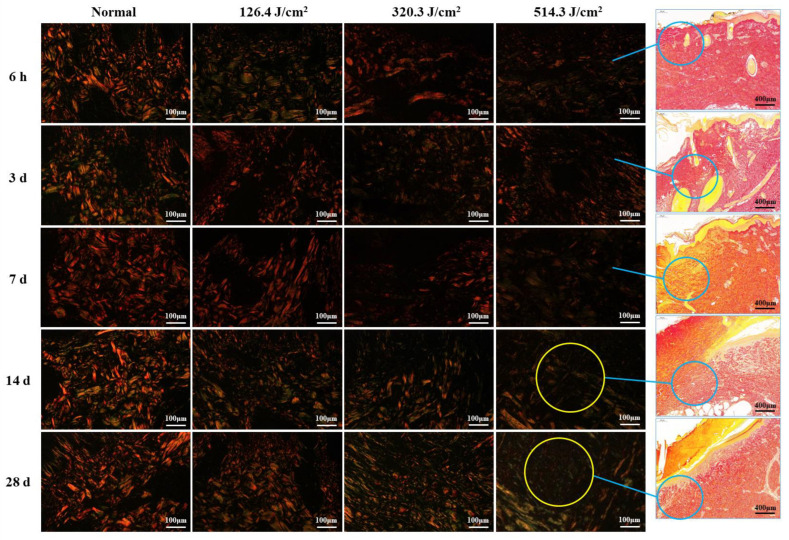
Evolution of collagen structure in skin wounds of miniature pigs post-laser irradiation (Sirius Red staining). Columns 1 through 4 represent images under a polarized light microscope, while column 5 presents images under a light microscope with varying scales. Blue circles indicate the approximate location of newly formed collagen fibers in the wounds, and yellow circles highlight the paucity of collagen fibers within the corresponding wounds (only numerous slender, immature, green type III collagen fibers).

**Figure 6 biology-13-00217-f006:**
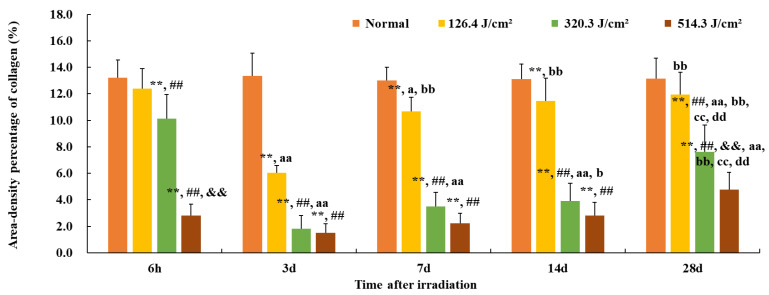
Quantitative analysis of area-density percentage of collagen in skin wounds 6 h to 28 d after laser irradiation. Error bars depict the standard deviation of the content of the different samples. Group comparisons are conducted through two-way ANOVA. In comparison to the normal group, ** *p* < 0.01; in comparison to low-dose group, ^##^
*p* < 0.01; in comparison to medium-dose group, ^&&^
*p* < 0.01; in comparison to the 6 h group, ^a^
*p* < 0.05, ^aa^
*p* < 0.01; in comparison to 3d group, ^b^
*p* < 0.05, ^bb^
*p* < 0.01; in comparison to 7d group, ^cc^
*p* < 0.01; in comparison to 14d group, ^dd^
*p* < 0.01.

**Figure 7 biology-13-00217-f007:**
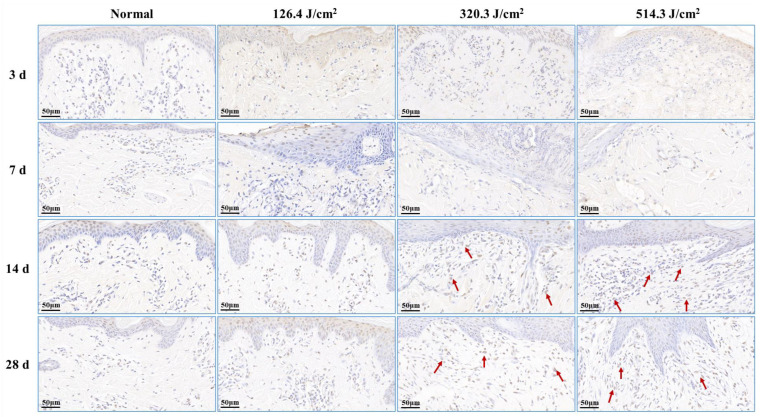
Alterations in apoptotic cells within pig skin wounds at various intervals following diverse doses of laser irradiation (TUNEL staining). Apoptotic cells in newly formed tissue are denoted by red arrows.

**Figure 8 biology-13-00217-f008:**
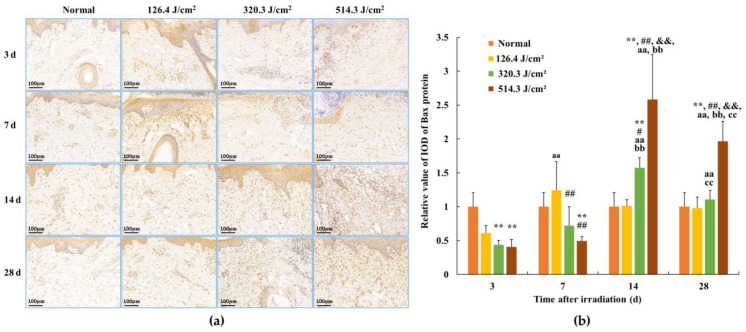
Alterations in Bax protein expression within pig skin wounds at different intervals following diverse doses of laser irradiation (IHC staining). Error bars depict the standard deviation of the expression of the different samples. Group comparisons were conducted through two-way ANOVA. (**a**) Positive protein expression observed through immunohistochemical staining; (**b**) quantitative analysis of Bax protein expression within skin wound tissue. In comparison to the normal group, ** *p* < 0.01; in comparison to low-dose group, ^#^
*p* < 0.05, ^##^
*p* < 0.01; in comparison to medium-dose group, ^&&^
*p* < 0.01; in comparison to the 3d group, ^aa^
*p* < 0.01; in comparison to 7d group, ^bb^
*p* < 0.01; in comparison to 14d group, ^cc^
*p* < 0.01.

**Figure 9 biology-13-00217-f009:**
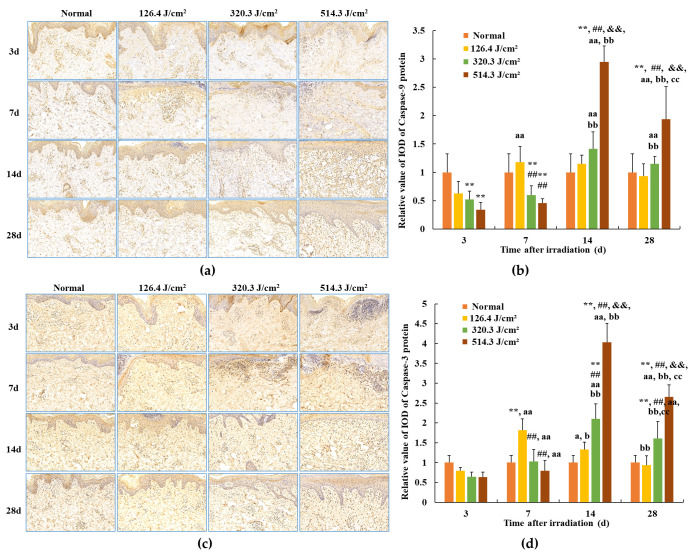
Modifications in caspase protein expression within pig skin wounds across distinct intervals following varied doses of laser irradiation (IHC staining). Error bars depict the standard deviation of the expression of the different samples. Group comparisons were conducted through two-way ANOVA. In comparison to the normal group, ** *p* < 0.01; in comparison to low-dose group, ^##^
*p* < 0.01; in comparison to medium-dose group, ^&&^
*p* < 0.01; in comparison to 3d group, ^a^
*p* < 0.05, ^aa^
*p* < 0.01; in comparison to 7d group, ^b^
*p* < 0.05, ^bb^
*p* < 0.01; in comparison to 14d group, ^cc^
*p* < 0.01. (**a**) Positive caspase-9 protein expression through iHC staining; (**b**) quantitative analysis of caspase-9 protein expression; (**c**) positive caspase-3 protein expression through IHC staining; (**d**) quantitative analysis of caspase-3 protein expression.

## Data Availability

All data are included in the manuscript. De-identified image data will be available upon request.

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
