# Peer review of "A Preliminary Study on Quantitative Analysis of Collagen and Apoptosis Related Protein on 1064 nm Laser-Induced Skin Injury"

_biology, 2024, doi:10.3390/biology13040217_

Round 1
Reviewer 1 Report (New Reviewer)
Comments and Suggestions for Authors
In the manuscript entitled, “Quantitative analysis of collagen and apoptosis related protein on 1064 nm laser-induced skin injury”, the authors show the mechanisms governing laser-induced skin injury and subsequent repair. The manuscript, particularly statistical analysis and figures need extensive improvements. All the data needs to be re-analyzed and all the figures need to be remade before the manuscript can be re-assessed for publication.
Comments:
1. Introduction is very long, and thus, the strengths of the study are lost on the readers. Please revise to make it concise, keeping only the important points for increasing readability.
2. The authors use two factors for the assessments in the current study: laser doses and time points. Please use Two Way ANOVA for determining the statistical significance of the differences of different groups at different time points as well as the comparison of the two factors and their interactions. Please revise the manuscripts, and all the figures, accordingly, highlighting the damage (structural and cell death) due to different laser intensities and wound healing as a function of time. Once Two Way ANOVA is performed, an interaction effect (p value) can also be included in the figures, in addition to the significance between the groups at each time point.
3. In the methods section, please mention the sex of the animals used and provide appropriate justifications for the same.
4. Please mention in the methods section if any age and sex matched control animals were used for the study.
5. Please provide appropriate quantitative analysis and statistics as well as control images for all the figures. Some figures have normal animal images; please mention how these images were obtained as no control animals have been described in the study. Please also mention in each figure legend what the error bars depict.
6. In Figure 2, the authors provide the details of how the images were marked for software analysis and mention that 5 sections were used for imaging. Please mention the detailed analysis process i.e. at what magnification and were averages calculated. It would also be useful if all the quantitative data be provided as supplementary figures.
7. Please mention if any quantitation (with appropriate statistical analysis) were done for the clinical assessments performed for these animals. Please include those in the manuscript, in support of Figure 3. Please also include control/normal images in Figure 3.
8. In Figure 4, only two time points have been shown and insets (enlarged) have been shown at only the study end point. Please include images and enlarged insets for all the time points are all the groups, including the control. This would help the reader appreciate effect of different laser doses on collagen damage and wound healing as a function of time. Please also perform quantitative assessments/statistical analysis and provide raw data as supplemental information. Additionally, the text in red in the last row is not legible and the pictures are not sharp enough to see clearly what the green arrows are indicating (new collagen); please improve figure quality.
9. Again, in Figure 5, include quantitative/statistical analyses. Please define what the collagen cavity indicates and how it was determined, using quantitative measurements. For example, less than X fibers indicated a collagen cavity. It is also not clear what the authors mean by “collagen structures” that are approximated by blue circles. Please provide these details in the text also.
10. In Figures 6, 8, and 9, please consider revising the clustering of the bar graphs. It would be much better to cluster time points and compare different groups in each cluster, as most comparisons are made between the normal and laser exposed groups.
11. In Figure 7, please add a quantitative analysis and zoom-in the images to clearly indicate the apoptotic/dead cells. They are not differentiable between the control and laser exposure groups in the current figures.
12. In Figure 8, there is only bar (for the 7 d time point) in the normal control; please explain this. In Figure 9, the normal/control group is completely missing in the graphs; please include.
13. Please considering including a conclusion figure, summarizing the major outcomes from the study.
Comments on the Quality of English LanguageMinor editing requireed.
Author Response
Thank you very much for taking the time to review this manuscript. Please find the detailed responses below. Due to time constraints for revision, minor editing of English language in this manuscript will be made after all comments have been properly resolved.
Comments 1: Introduction is very long, and thus, the strengths of the study are lost on the readers. Please revise to make it concise, keeping only the important points for increasing readability.
Response 1: I have simplified the introduction.
Comments 2: The authors use two factors for the assessments in the current study: laser doses and time points. Please use Two Way ANOVA for determining the statistical significance of the differences of different groups at different time points as well as the comparison of the two factors and their interactions. Please revise the manuscripts, and all the figures, accordingly, highlighting the damage (structural and cell death) due to different laser intensities and wound healing as a function of time. Once Two Way ANOVA is performed, an interaction effect (p value) can also be included in the figures, in addition to the significance between the groups at each time point.
Response 2: This study focused on the changes in collagen content and apoptotic protein expression during the healing process of laser-induced skin injury, highlighting the effect of laser dose on laser-induced skin injury and healing and weakening the interactive effect of two factors, laser dose and time point. One-way ANOVA was mostly used in many related studies; if two-way ANOVA is necessary, I will refine it.
Please review lines 214-218 in the manuscript.
Comments 3: In the methods section, please mention the sex of the animals used and provide appropriate justifications for the same.
Response 3: Five miniature pigs of either sex (3 females and 2 males) was used in this experiment. We found that animal sex had no effect on studies of skin damage.
Please review line 113 in the manuscript.
Comments 4: Please mention in the methods section if any age and sex matched control animals were used for the study.
Response 4: No age- and sex-matched control animals were used in this study. The normal skin sample of each animal at the unirradiated site was used as a control in this study.
Please review lines 158-159 in the manuscript.
Comments 5: Please provide appropriate quantitative analysis and statistics as well as control images for all the figures. Some figures have normal animal images; please mention how these images were obtained as no control animals have been described in the study. Please also mention in each figure legend what the error bars depict.
Response 5: Normal skin from the unirradiated area of each animal was used as a control and was taken for this study. The normal images were obtained from the above-mentioned samples. Error bars depict the standard deviation of the content or expression of the different samples. I have added the representation of error bars in each legend.
Please review Figures 6, 8, and 9 in the manuscript.
Comments 6: In Figure 2, the authors provide the details of how the images were marked for software analysis and mention that 5 sections were used for imaging. Please mention the detailed analysis process i.e. at what magnification and were averages calculated. It would also be useful if all the quantitative data be provided as supplementary figures.
Response 6: The IOD values for these images (magnification of 400) were obtained by Image-Pro Plus 6.0 software, as shown in the figure below, and then the mean and standard deviation were calculated using SPSS 13.0 software. Detailed data will be provided as an attachment at the end of this document.
Please review lines 201-205 in the manuscript.
Comments 7: Please mention if any quantitation (with appropriate statistical analysis) were done for the clinical assessments performed for these animals. Please include those in the manuscript, in support of Figure 3. Please also include control/normal images in Figure 3.
Response 7: In this study, the camera was used to capture images of the entire skin wound region, and all wounds were not in the same plane, which affected the area of the wounds at different shooting angles. Therefore, we have not found a suitable method of quantitative analysis, and only qualitative analysis was performed in the manuscript. Normal group images have been added in Figure 3.
Please review Figure 3 in the manuscript.
Comments 8: In Figure 4, only two time points have been shown and insets (enlarged) have been shown at only the study end point. Please include images and enlarged insets for all the time points are all the groups, including the control. This would help the reader appreciate effect of different laser doses on collagen damage and wound healing as a function of time. Please also perform quantitative assessments/statistical analysis and provide raw data as supplemental information. Additionally, the text in red in the last row is not legible and the pictures are not sharp enough to see clearly what the green arrows are indicating (new collagen); please improve figure quality.
Response 8: This part of the work have been discussed in detail in another published article (FanY and Ma Q et al. Quantitative and qualitative evaluation of recovery process of a 1064nm laser on laser-induced skin injury: in vivo experimental research. Laser Phys Lett 2019; 16, 115604). In this study, pathological observations of skin wound healing at 6h and 3d post-irradiation are shown as a brief prelude to subsequent work. Image-related comments have been revised as requested by the reviewers.
Please review Figure 4 in the manuscript.
Comments 9: Again, in Figure 5, include quantitative/statistical analyses. Please define what the collagen cavity indicates and how it was determined, using quantitative measurements. For example, less than X fibers indicated a collagen cavity. It is also not clear what the authors mean by “collagen structures” that are approximated by blue circles. Please provide these details in the text also.
Response 9: Blue circles indicate the approximate location of newly formed collagen fibers in the wounds, and yellow circles highlight the paucity of collagen fibers within the corresponding wounds (only numerous immature, slender, green type III collagen fibers). The quantitative/statistical analyses of collagen in low-, medium-, and high-dose groups at 6h~28d post-irradiation were shown in Figure 6.
Please review Figure 5 in the manuscript.
Comments 10: In Figures 6, 8, and 9, please consider revising the clustering of the bar graphs. It would be much better to cluster time points and compare different groups in each cluster, as most comparisons are made between the normal and laser exposed groups.
Response 10: Figures 6, 8, and 9 have been revised as temporal clustering plots.
Comments 11: In Figure 7, please add a quantitative analysis and zoom-in the images to clearly indicate the apoptotic/dead cells. They are not differentiable between the control and laser exposure groups in the current figures.
Response 11: We also attempted to perform manual counts of apoptotic/dead cells, but due to the large amount of work and the discrepancies caused by subjectivity, this study is based on the qualitative description of the presence of varying numbers of apoptotic cells. We will carry out an in-depth study on automated counting of apoptotic and necrotic cells in conjunction with deep learning on the results of this part of the experiment.
Comments 12: In Figure 8, there is only bar (for the 7 d time point) in the normal control; please explain this. In Figure 9, the normal/control group is completely missing in the graphs; please include.
Response 12: Figure 8 and 9 has been revised in accordance with reviewer’s comments. The expression of normal skin was used as a control.
Comments 13: Please considering including a conclusion figure, summarizing the major outcomes from the study.
Response 13: This study focuses on a preliminary investigation of two important factors (insufficient collagen synthesis and excessive cellular apoptosis) that contribute to the delayed healing of skin injuries induced by medium- to high-dose lasers. The conclusions mentioned these two factors and the interrelationships are relatively brief, and I feel that a conclusion figure may create a repetitive description and does not provide the reader with additional information; if it is necessary, I will refine it.
Attachment
- The corresponding figure in the manuscript is Figure 8.
Schedule 1-1 Quantitative analysis of Bax protein expression in pig skin wounds at different times after different doses of laser irradiation (IOD values)
|
Time after irradiation (d) |
Normal |
Laser irradiation dose (J/cm2) |
||
|
126.4 |
320.3 |
514.3 |
||
|
3 |
16625 ± 3482.8 |
10117 ± 1900.8** |
7238 ± 1129.5**, ## |
6777± 1810.4**, ## |
|
7 |
20685 ± 7025.6 |
11946 ± 4701.8*, ## |
8255 ± 1055.9**, ##, & |
|
|
14 |
16894 ± 1410.3 |
26134 ± 2524.8**, ## |
42935 ± 11072.4**, ##, && |
|
|
28 |
16292 ± 2709.6 |
18415 ± 2191.0 |
32676 ± 4933.4**, ##, && |
|
Note: Compared with the normal group, *P<0.05, **P<0.01; In comparison to low-dose group, #P<0.05, ##P<0.01ï¼›In comparison to medium-dose group, &P<0.05, &&P<0.01.
Schedule 1-2 Quantitative analysis of Bax protein expression in pig skin wounds at different times after different doses of laser irradiation (relative IOD values)
|
Time after irradiation (d) |
Normal |
Laser irradiation dose (J/cm2) |
||
|
126.4 |
320.3 |
514.3 |
||
|
3 |
1.0 ± 0.21 |
0.61± 0.11** |
0.44 ± 0.07**, ## |
0.41 ± 0.11**, ## |
|
7 |
1.24 ± 0.42 |
0.72 ± 0.28*, ## |
0.50 ± 0.06**, ##, & |
|
|
14 |
1.02 ± 0.08 |
1.57 ± 0.15**, ## |
2.58 ± 0.67**, ##, && |
|
|
28 |
0.98 ± 0.16 |
1.11 ± 0.13 |
1.97 ± 0.30**, ##, && |
|
Note: Compared with the normal group, *P<0.05, **P<0.01; ; In comparison to low-dose group, #P<0.05, ##P<0.01ï¼›In comparison to medium-dose group, &P<0.05, &&P<0.01.
- The corresponding figure in the manuscript is Figure 9.
Schedule 2-1-1 Quantitative analysis of caspase-9 protein expression in pig skin wounds at different times after different doses of laser irradiation (IOD values)
|
Time after irradiation (d) |
Normal |
Laser irradiation dose (J/cm2) |
||
|
126.4 |
320.3 |
514.3 |
||
|
3 |
20424 ± 6634.1 |
12847 ± 4271.6* |
10608 ± 3067.3** |
7004± 2540.1**, ##, && |
|
7 |
24107 ± 5636.8 |
12212 ± 3390.8**, ## |
9350 ± 1501.5**, ##, & |
|
|
14 |
23547 ± 3084.6 |
28928 ± 6108.5* |
60253 ± 5751.1**, ##, && |
|
|
28 |
19079 ± 4437.3 |
23443 ± 2698.3# |
39554 ± 11828.3**, ##, && |
|
Note: Compared with the normal group, *P<0.05, **P<0.01; ; In comparison to low-dose group, #P<0.05, ##P<0.01ï¼›In comparison to medium-dose group, &P<0.05, &&P<0.01.
Schedule 2-1-2 Quantitative analysis of caspase-9 protein expression in pig skin wounds at different times after different doses of laser irradiation (relative IOD values)
|
Time after irradiation (d) |
Normal |
Laser irradiation dose (J/cm2) |
||
|
126.4 |
320.3 |
514.3 |
||
|
3 |
1.00 ± 0.325 |
0.63 ± 0.209* |
0.52 ± 0.150** |
0.34 ± 0.124**, ##, && |
|
7 |
1.18 ± 0.276 |
0.60 ± 0.166**, ## |
0.46 ± 0.074**, ##, & |
|
|
14 |
1.15 ± 0.151 |
1.42 ± 0.299* |
2.95 ± 0.282**, ##, && |
|
|
28 |
0.93 ± 0.217 |
1.15 ± 0.132# |
1.94 ± 0.579**, ##, && |
|
Note: Compared with the normal group, *P<0.05, **P<0.01; ; In comparison to low-dose group, #P<0.05, ##P<0.01ï¼›In comparison to medium-dose group, &P<0.05, &&P<0.01.
Schedule 2-2-1 Quantitative analysis of caspase-3 protein expression in pig skin wounds at different times after different doses of laser irradiation (IOD values)
|
Time after irradiation (d) |
Normal |
Laser irradiation dose (J/cm2) |
||
|
126.4 |
320.3 |
514.3 |
||
|
3 |
18400 ± 3288.3 |
14600 ± 1613.3* |
11793 ± 2272.3**, # |
11641 ± 2341.1**, # |
|
7 |
33371 ± 5383.8** |
18880 ±5606.4## |
14665 ± 4759.5*, ## |
|
|
14 |
24435 ± 3413.4** |
38748 ± 6888.1**, ## |
74238 ± 8681.6**, ##, && |
|
|
28 |
17285 ± 4351.7 |
29628 ± 7901.3**, ## |
48949 ± 5484.3**, ##, && |
|
Note: Compared with the normal group, *P<0.05, **P<0.01; ; In comparison to low-dose group, #P<0.05, ##P<0.01ï¼›In comparison to medium-dose group, &P<0.05, &&P<0.01.
Schedule 2-2-2 Quantitative analysis of caspase-3 protein expression in pig skin wounds at different times after different doses of laser irradiation (relative IOD values)
|
Time after irradiation (d) |
Normal |
Laser irradiation dose (J/cm2) |
||
|
126.4 |
320.3 |
514.3 |
||
|
3 |
1.00 ± 0.179 |
0.79 ± 0.088* |
0.64 ± 0.123**, # |
0.63 ± 0.127**, # |
|
7 |
1.81 ± 0.293** |
1.03 ± 0.305## |
0.80 ± 0.259*, ## |
|
|
14 |
1.33 ± 0.186** |
2.11 ± 0.374**, ## |
4.03 ± 0.472**, ##, && |
|
|
28 |
0.94 ± 0.237 |
1.61 ± 0.429**, ## |
2.66 ± 0.298**, ##, && |
|
Note: Compared with the normal group, *P<0.05, **P<0.01; ; In comparison to low-dose group, #P<0.05, ##P<0.01ï¼›In comparison to medium-dose group, &P<0.05, &&P<0.01.

Reviewer 2 Report (New Reviewer)
Comments and Suggestions for Authors
Here is my opinion about the work: Quantitative analysis of collagen and apoptosis related protein on 1064 nm laser-induced skin injury
Lasers find widespread use in our life, in medical procedures, industrial settings, communication technologies or scientific research. Despite such wide use, knowledge about the biological impact of lasers is still insufficient. The work described the effect of 1064 nm laser irradiation on skin cells in a porcine model, whose skin healing process is very similar to that of humans. The work is generally well-written, but the major limitations of the work are a low number of animals used in the experiment (1 per time point), and the relatively low number of analyses (collagens, apoptosis with apoptotic proteins). Therefore, authors should clearly indicate in the abstract and maybe even in the title that this is a preliminary study performed on 1 individual at 1-time point, and in this respect, the final conclusions should also be written.
Here changes that should be done before publish the paper.
1. Simple Sumarry:
- The cell types of laser-induced apoptosis in porcine skin injuries mainly encompassed fibroblasts, vascular endothelial cells, and epidermal cells. - Authors did not examine cells types in wound - the sentence should be deleted or written according to what was examined
2. Abstract:
- pleases add one general sentence about how lasers work.
- authors did not explore damage in
3. Introduction:
- lines 64-67 please add references
- lines 74-75 - remove: “Laser weapons may also cause irreversible damage to the skin of personnel when attacking a target, and similarly” - it is unnecessary here, leave the rest of the sentence
- lines 94-96 should be rewritten - the influx of immune system cells itself does not change the arrangement of proteins - this happens mainly due to the active action of inflammatory immune cells and the secretion of appropriate compounds by them. Additionally, it is worth remembering that some cells of the immune system (i.e. MQ M2 or Treg/Breg cells) play a remedial role by also secreting a factor that supports cell proliferation.
- lines: 110-111 delete: and it is hypothesized that cell proliferation and apoptosis play an important role in this process, - it is well known that in wound healing both processes play a pivotal role, what was described above and below.
- lines: 125-130 delete both sentences. The study performed on 5 Guizhou miniature fragrant pigs (one per appropriate time point!!) is a very preliminary study - the observed relationships may result from individual wound healing conditions and not from the assumed mechanism.
4. M&M section:
- 173-4: This avoided continuous observation of the same animal over the full 28-day span. - This is a rather unfavorable approach from the perspective of examining the wound healing process.
- 2.3 Method to observe damage spots - the irradiation spot's diameter (d) was standardized at 1 cm - it should be clearly described how spot's irradiation sin surface was detected for histopathological analysis.
- in histopathological scraps - the effect of immune cell infiltration analysis should be added to the work. This would increase the quality of the work submitted for evaluation
- 2.4 -6: There is no description how authors prepared samples for analysis - it should be added (concentration, time, reagents, temp. etc)
- 2.7 Statistical analysis - there is no information on which ANOVA (parametric or not parametric) and which post-test were used.
5. Results section:
- Figure 3 - line 255 - … and green arrows denote - add large
- Figure 4. - Please add magnification and microscope name.
- line 335 : please define why: apoptotic cells in skin wounds at 6h to 3d post-irradiation are not depicted in Figure 7.? and add these pictures to figure 7. Moreover, add to Figure 7 numerical graphs of the % of apoptosis and necrosis recorded in the histopathological analysis of samples
6. Discussion:
- authors should clearly indicate in the abstract and maybe even in the title that this is a preliminary study performed on 1 individual at 1-time point, and in this respect, the final conclusions should also be written.
- 514-547: The authors presented a hypothesis why apoptosis at the highest concentrations is 5 times higher than after 3 days, compared to 14 and 28 days but they don’t discuss it with other works. The paragraph should be rewritten.
7. Conclusions: this is a preliminary study performed on 1 individual at 1-time point, and in this respect, the final conclusions should also be written.
Author Response
Thank you very much for taking the time to review this manuscript. Please find the detailed responses below.
Comments 1(Simple Sumarry):
-The cell types of laser-induced apoptosis in porcine skin injuries mainly encompassed fibroblasts, vascular endothelial cells, and epidermal cells. - Authors did not examine cells types in wound - the sentence should be deleted or written according to what was examined.
Response: The description of “The cell types of laser-induced apoptosis in porcine skin injuries mainly encompassed fibroblasts, vascular endothelial cells, and epidermal cells” has been removed from my manuscript. It is simply described as “Laser-induced apoptotic cells in porcine skin injuries were significantly increased in the late stages of healing.”
Please review lines 12-15 in the manuscript.
Comments 2(Abstract):
-pleases add one general sentence about how lasers work. -authors did not explore damage in.
Response: Live pig skin was irradiated at multiple spots one time by using a grid-array method with a 1064 nm laser at different power outputs. The skin injuries primarily manifested as white spots and burnt spots.
Please review lines 19-21 in the manuscript.
Comments 3(Introduction):
-lines 64-67 please add references
Response: I have added relevant references to the manuscript.
- Mehrabi JN, Kelly KM, Holmes JD, Zachary CB. Assessing the Outcomes of Focused Heating of the Skin by a Long-Pulsed 1064 nm Laser with an Integrated Scanner, Infrared Thermal Guidance, and Optical Coherence Tomography. Lasers Surg Med. 2021 Aug; 53(6):806-814.
- Leight-Dunn H, Chima M, Hoss E. Wound Healing Treatments After Ablative Laser Skin Resurfacing: A Review. J Drugs Dermatol. 2020 Nov 1; 19(11):1050-1055.
-lines 74-75 - remove: “Laser weapons may also cause irreversible damage to the skin of personnel when attacking a target, and similarly” - it is unnecessary here, leave the rest of the sentence
Response: The description of " Laser weapons may also cause irreversible damage to the skin of personnel when attacking a target, and similarly " has been removed from my manuscript.
-lines 94-96 should be rewritten - the influx of immune system cells itself does not change the arrangement of proteins - this happens mainly due to the active action of inflammatory immune cells and the secretion of appropriate compounds by them. Additionally, it is worth remembering that some cells of the immune system (i.e. MQ M2 or Treg/Breg cells) play a remedial role by also secreting a factor that supports cell proliferation.
Response: I agree with your viewpoint. This section has been drastically deleted for the sake of brevity and clarity of introduction. The description of " Nevertheless, as the duration of skin injury prolongs, inflammatory cells persistently infiltrate the wound site. This shift in the protein milieu culminates in the continuous degradation of growth factors, attenuating their capacity to foster healing, consequently inducing wound healing delay” has been removed from my manuscript.
-lines: 110-111 delete: and it is hypothesized that cell proliferation and apoptosis play an important role in this process, - it is well known that in wound healing both processes play a pivotal role, what was described above and below.
Response: The description of "and it is hypothesized that cell proliferation and apoptosis play an important role in this process" has been removed from my manuscript.
-lines: 125-130 delete both sentences. The study performed on 5 Guizhou miniature fragrant pigs (one per appropriate time point!!) is a very preliminary study - the observed relationships may result from individual wound healing conditions and not from the assumed mechanism.
Response: “This investigation imparts an experimental and theoretical foundation for comprehending the ramifications and mechanisms of cutaneous thermal injuries triggered by a 1064 nm infrared laser. Additionally, it furnishes an indispensable quantitative and qualitative re-search approach for the future exploration of laser-induced skin wound effects and the recovery process” has been removed from my manuscript. It is simply described as " This study provides experimental ideas and methods for future exploration of the consequences and mechanisms of cutaneous thermal injuries triggered by a1064 nm infrared laser."
Please review lines 108-110 in the manuscript.
Comments 4(M&M):
-173-4: This avoided continuous observation of the same animal over the full 28-day span. - This is a rather unfavorable approach from the perspective of examining the wound healing process.
Response: I agree with the limitations of this assay. However, this method reduces the chance of results that may result from observing only 1 animal. Additionally, the experiment used a comparative analysis with the initial injury situation to observe the effect of individual differences on injury and healing.
Please review lines 148-156 in the manuscript.
-2.3 Method to observe damage spots - the irradiation spot's diameter (d) was standardized at 1 cm - it should be clearly described how spot's irradiation sin surface was detected for histopathological analysis.
Response: Exudation, bleeding, crusting or debridement of the skin wound surface is recorded to provide a reference for histopathological analysis. One pig was humanely euthanized to obtain samples at a particular time point, and the wound area is sampled along with the surrounding normal tissue.
Please review lines 148-156 in the manuscript.
-in histopathological scraps - the effect of immune cell infiltration analysis should be added to the work. This would increase the quality of the work submitted for evaluation
Response: This is a great suggestion. Wound samples for histopathological analysis were all fixed in 4% neutral formaldehyde solution as soon as they were obtained, without histopathological scraps. This assay will be added to future experiments.
-2.4 -6: There is no description how authors prepared samples for analysis - it should be added (concentration, time, reagents, temp. etc)
Response: The wound samples were fixed in 4% neutral formaldehyde solution, and these pre-treated samples were sent to Wuhan Servicebio Biotechnology Company for paraffin embedding, sectioning, and staining. The company provides a standardized procedure with specific experimental parameters for each step (concentration, time, reagents, temp. etc).
Please review lines 160-164 in the manuscript.
-2.7 Statistical analysis - there is no information on which ANOVA (parametric or not parametric) and which post-test were used.
Response: Group comparisons were conducted through one-way ANOVA. For further two-by-two comparisons, LSD-t test was used when the variances were aligned, and Dunnett's T3 method was used when the variances were not aligned. Statistical significance was set at P < 0.05.
Please review lines 214-218 in the manuscript.
Comments 5(Results):
- Figure 3 - line 255 - … and green arrows denote - add large
Response: Green arrows denote large burnt spots.
Please review line 245 in the manuscript.
-Figure 4. - Please add magnification and microscope name.
Response: Pathology images were acquired using a digital pathology system (Pannoramic MIDI, 3DHISTECH Int.). The magnification has been added to my manuscript. The images in the third row correspond to magnified views marked by boxes in the second row, and the magnification is 25.
Please review lines 164-166 in the manuscript.
-line 335 : please define why: apoptotic cells in skin wounds at 6h to 3d post-irradiation are not depicted in Figure 7.? and add these pictures to figure 7. Moreover, add to Figure 7 numerical graphs of the % of apoptosis and necrosis recorded in the histopathological analysis of samples
Response: The skin wounds were dominated by the damage response, with minimal neoplastic tissue and apoptotic cells in the skin wounds at 6 h and 3d post-irradiation. To better understand the trend of apoptotic cell changes, images of apoptotic cells 3d post-irradiation were added in Figure 7. We also attempted to perform manual counts of apoptotic and necrotic cells, but due to the large amount of work and the discrepancies caused by subjectivity, this study is based on the qualitative description of the presence of varying numbers of apoptotic cells. We will carry out an in-depth study on automated counting of apoptotic and necrotic cells in conjunction with deep learning on the results of this part of the experiment.
Please review Figure 7 in the manuscript.
Comments 6(Discussion):
-authors should clearly indicate in the abstract and maybe even in the title that this is a preliminary study performed on 1 individual at 1-time point, and in this respect, the final conclusions should also be written.
Response: Considering animal welfare issues, the experimental protocol of this study used a minimum number of 5 live animals, and only one pig was humanely euthanized at a particular time point from 6 hours to 28 days post-irradiation. Based on the reviewers' suggestions, I have revised the abstract, title and conclusion accordingly. This study only explores the important factors influencing the healing of laser-induced skin injuries without mentioning the mechanisms.
-514-547: The authors presented a hypothesis why apoptosis at the highest concentrations is 5 times higher than after 3 days, compared to 14 and 28 days but they don’t discuss it with other works. The paragraph should be rewritten.
Response: I don't really understand the reviewer's comment. There is no mention of such a hypothesis in the manuscript. Figures 6, 8, and 9 have been revised as temporal clustering plots, and and a statistical analysis between the different groups was added.
Please review Figures 6, 8, and 9 in the manuscript.
Comments 7(Conclusions):
-this is a preliminary study performed on 1 individual at 1-time point, and in this respect, the final conclusions should also be written.
Response: Based on the reviewers' suggestions, I have revised the abstract, title and conclusion accordingly. This study focuses on a preliminary investigation of two important factors (insufficient collagen synthesis and excessive cellular apoptosis) that contribute to the delayed healing of skin injuries induced by medium- to high-dose lasers.
Reviewer 3 Report (New Reviewer)
Comments and Suggestions for Authors
The authors did a comprehensive study on wound healing of 1064nm laser induced skin injury in miniature pig model and identified the alterations of collagen and apoptosis-related proteins associated with wound healing after the injury. The detailed review comments are listed below.
Line 110: “a1064” should change to “a 1064”.
Line 215: ANOVA is in the section 2.7 Statistical analysis, but not clear which figure(s) used ANOVA. Please label it in the figure legends or include the information in the section 2.7 if applicable.
Line 245: how to define small/large burnt spots, by area or diameter of the wound sites?
Line 268: the blue circle cannot show the accurate wound sites considering the depth and other aspects, some explanation would help, or change to bars to show the left and right boundaries.
Lines 268-269: the sentence “The images in the third row 268 correspond to magnified views marked by boxes in the second row.” is the same as in lines 267-268, should be deleted.
Line 323: for the low-dose group, what is the % of the normal group?
Line 374: “norma” should change to “normal”.
Comments on the Quality of English LanguageOverall the writing is OK. Some minor parts could be improved, for example the "witnessed" in line 405 could change to "showed", or "XXX was observed in XXX groups".
Author Response
Thank you very much for taking the time to review this manuscript. Please find the detailed responses below.
Line 110: “a1064” should change to “a 1064”.
Response: It has been corrected in the manuscript.
Line 215: ANOVA is in the section 2.7 Statistical analysis, but not clear which figure(s) used ANOVA. Please label it in the figure legends or include the information in the section 2.7 if applicable.
Response: Group comparisons are conducted through one-way ANOVA. This has been label in the figure legends, including Figure 6, 8, and 9.
Line 245: how to define small/large burnt spots, by area or diameter of the wound sites?
Response: The small/large burnt spots are not defined by area or diameter of the wound sites. As shown in Figure 3, blue arrows denote small burnt spots at the center of white spots, and green arrows denote large burnt spots.
Line 268: the blue circle cannot show the accurate wound sites considering the depth and other aspects, some explanation would help, or change to bars to show the left and right boundaries.
Response: The blue box has replaced the blue circle to show the left and right boundaries of wound sites. Please review Figure 4 in the manuscript.
Lines 268-269: the sentence “The images in the third row 268 correspond to magnified views marked by boxes in the second row.” is the same as in lines 267-268, should be deleted.
Response: “The images in the third row 268 correspond to magnified views marked by boxes in the second row” has been deleted in the manuscript.
Line 323: for the low-dose group, what is the % of the normal group?
Response: The area-density percentage of collagen in the low-group is (11.9 ± 1.7) % (91% of the normal gruop). This result has been added in the manuscript.
Line 374: “norma” should change to “normal”.
Response: The " norma " in line 374 has been changed to "normal".
Comments on the Quality of English Language Overall the writing is OK. Some minor parts could be improved, for example the "witnessed" in line 405 could change to "showed", or "XXX was observed in XXX groups".
Response: The "witnessed" in line 405 has been changed to "showed".

Round 2
Reviewer 1 Report (New Reviewer)
Comments and Suggestions for Authors
The authors have revised the manuscript; however, it still needs much improvement. The response #8 is very confusing; did the authors use the image/data (figure 4) that has already been published in a different study, very confusing (and alarming). In this study, molecular (collagen) expression is being analyzed as a function of time (interaction of two factors). Additionally, both test and control samples have been obtained for the same animal. All these factors must be accommodated for while performing the statistical analysis. Thus, the result and figures are not acceptable in the current form. Overall, manuscript has scope but after revision it still needs major correction. Please reconsider re-analyzing the data and provide all the control Figures as well as quantitative analysis, which are missing in the manuscript. The authors should remove the results for which they are not able to perform quantitative analysis, as the differences between the groups cannot be assessed based on only a few qualitative observations. I am unable to recommend the present manuscript for publication.
Author Response
Thank you very much for taking the time to review this manuscript. Please find the detailed responses below.
The response #8 is very confusing; did the authors use the image/data (figure 4) that has already been published in a different study, very confusing (and alarming).
Response: Morphological examination of pig skin injury and healing from 6 hours to 28 days post-irradiation has been published in another article (FanY and Ma Q et al. Quantitative and qualitative evaluation of recovery process of a 1064nm laser on laser-induced skin injury: in vivo experimental research. Laser Phys Lett 2019; 16, 115604). The article exhaustively analysed the morphological changes in laser-induced skin injuries during the healing process and comparatively analysed the statistical differences between the groups, as shown in the figure below (text and images taken from the original article).
In this article, the main purpose of introducing the 6h and 24h histomorphological results is to take up the macroscopic observations and pave the way for the subsequent collagen and immunohistological staining results, thus increasing the readability of the article. Based on the reviewers' comments, this section has been revised. Specifically, morphological analyses were performed for each group, without comparative analyses between groups. Please review lines 245-260 in the manuscript.
In this study, molecular (collagen) expression is being analyzed as a function of time (interaction of two factors). Additionally, both test and control samples have been obtained for the same animal. All these factors must be accommodated for while performing the statistical analysis. Thus, the result and figures are not acceptable in the current form. Overall, manuscript has scope but after revision it still needs major correction. Please reconsider re-analyzing the data and provide all the control Figures as well as quantitative analysis, which are missing in the manuscript.
Response: Data in the manuscript were reanalysed in response to reviewer comments, as shown in Schedule 1-3. The results were subjected to analysis using GraphPad Prism 9.0 software. Group comparisons were conducted through two-way ANOVA (Tukey test).
The authors should remove the results for which they are not able to perform quantitative analysis, as the differences between the groups cannot be assessed based on only a few qualitative observations. I am unable to recommend the present manuscript for publication.
Response: I agree with your views on quantitative and qualitative analysis. The main purpose of apoptosis analysis is to observe the changes of apoptotic cells in laser-induced skin injuries during the healing process. Based on the reviewers' comments, apoptosis analyses were performed for each group, without comparative analyses between groups. Please review lines 339-345 in the manuscript.

Reviewer 2 Report (New Reviewer)
Comments and Suggestions for Authors
the authors responded satisfactorily to my comments and introduced the corrected text into the work. In my opinion, the work can be published in this form.
Author Response
Thank you very much for taking the time to review this manuscript.
This manuscript is a resubmission of an earlier submission. The following is a list of the peer review reports and author responses from that submission.
Round 1
Reviewer 1 Report
Comments and Suggestions for Authors
Manuscript biology-2746953 “Quantitative analysis of collagen and apoptosis related protein of a 1064 nm laser on laser-induced skin injury” is a research paper evaluating the effect of 1064 nm laser on pig skin. The authors assessed the effects of different intensities of laser irradiation on collagen deposition and apoptotic protein expression at different time points, and the results of histological analysis indicate that a middle to high dose of the laser impair collagen deposition and promote apoptosis during wound-healing process. The objectives, methods, and results were clear and no major problems were identified.
Please describe the availability of equipment and reagents. Part numbers should also be listed, especially for antibodies.
Lines 178-179, "Beijing Boaosenthesis Biotechnology" might be corrected to Beijing Biosynthesis Biotechnology."
Author Response
Thank you very much for taking the time to review this manuscript. Please find the detailed responses below.
Comments 1: Please describe the availability of equipment and reagents. Part numbers should also be listed, especially for antibodies.
Response 2: The 1064nm laser is a self-developed laser system that programmatically sets the output current and radiation time through a controller, and laser power is gauged using a laser power meter (BGS6333-P400, Beijing Monochrome Optoelectronic Technology Co., Ltd) before radiation. Components including Protease K (G1234), TUNEL kit (G1507), DAB chromogenic agent (G1212), and hema-toxylin staining solution (G1076) were acquired from Servicebio. Rabbit anti-mouse polyclonal antibodies for bax (bs-0127R), caspase-3 (bs-0081R), and caspase-9 (bs-20773R) were sourced from Beijing Boaosen Biotechnology Co., Ltd.
Comments 2: Lines 178-179, "Beijing Boaosenthesis Biotechnology" might be corrected to Beijing Biosynthesis Biotechnology."
Response 2. I have corrected to Beijing Boaosen Biotechnology Co., Ltd. This is a company engaged in biotechnology in Beijing.
Reviewer 2 Report
Comments and Suggestions for Authors
I could not understood the purpose of this study.
If a convincing background is provided in the Introduction and Discussion section, I think this paper is worthy of publication.
(1) The title should be changed to "Quantitative analysis of collagen and apoptosis related protein on 1064 nm laser-induced skin injury".
(2) What is the Authors' most important message to readers? Is the 1064 nm laser harmful? Why did the Authors choose the subject of skin damage caused by the 1064nm laser? Is the 1064nm laser used in clinics? The significance of this paper should be more clear.
(3) If a 1064nm laser is already in clinical use and the author believes its skin-damaging properties are problematic, background should be added in the Introduction section.
Author Response
Thank you very much for taking the time to review this manuscript. Please find the detailed responses below.
Comments 1: The title should be changed to "Quantitative analysis of collagen and apoptosis related protein on 1064 nm laser-induced skin injury".
Response 1: Agree. I have corrected it.
Comments 2: What is the Authors' most important message to readers? Is the 1064 nm laser harmful? Why did the Authors choose the subject of skin damage caused by the 1064nm laser? Is the 1064nm laser used in clinics? The significance of this paper should be more clear.
Response 2:
Accidental irradiation of the skin with higher doses of laser resulted in injuries of white coagulation spots and charred spots, however, the main factors affecting the delayed wound recovery of these lesions included the augmented apoptotic cell population and insufficient collagen production in the newborn skin tissue.
The 1064nm laser is widely used in the basic research, such as laser surgery, laser ther-motherapy and laser imaging. The 1064nm laser has a good deep-reaching capability and is widely used clinically for the treatment of skin disorders, such as non-ablative treatments for skin rejuvenation and ablative treatments for dermatologic diseases, and the harmful effects of lasers are mainly due to improper or excessive use; however, the high-power 1064nm laser is widely used in industrial and military applications, and its use may result in accidental skin burns to personnel.
The choice of 1064 nm laser skin injury as a research topic is firstly due to the wide range of applications of 1064 nm lasers; secondly, the skin is the main target organ for laser injuries except for the eyes; and thirdly, 1064 nm infrared laser is not visible by the naked eye thus leading to easier accidental injury.
Comments 3: If a 1064nm laser is already in clinical use and the author believes its skin-damaging properties are problematic, background should be added in the Introduction section.
Response 3:
The 1064nm laser has been widely used in clinical practice, and skin injuries caused by improper or excessive use as well as by high power irradiation are the focus of this study, and the relevant background has been revised in the manuscript. Some differences between laser irradiation and thermal burn in terms of skin damage and wound healing have been reported in the literature, which will be an important direction for conducting future studies.
Reviewer 3 Report
Comments and Suggestions for Authors
In my point of view the animal utilization for this study is unaceptable.
Despite the good results and the presentation of the data, I cannot accept that the research was carried out on live pigs in 2023. Despite approval by the ethics committee, we need to start replacing the ways we think about research. The same research could have been carried out even with pigskin left over from the butcher's shop. We already have great options to avoid using live animals in research. I do not feel able to review this type of research, as it is almost 2024. The study did not aim to observe the safety of the laser - which is already known - and the findings are not very different from what we know in the literature.
Author Response
Thank you very much for taking the time to review this manuscript. Please find the detailed responses below.
Response: I really appreciate and admire your rigorous attitude towards the welfare of laboratory animals. If only observing the immediate damage to the skin tissue after irradiation with a 1064nm laser, I think it is quite feasible to use fresh pork skin from the market. Isolated tissue, unlike living tissue, does not heal itself. In this study, we dynamically analyzed collagen levels and apoptosis-related protein expressions during the healing process of porcine skin injuries induced by laser irradiation. Therefore, due to the great recognition of the welfare of animals, the experimental protocol of this study used a minimum number of 5 live animals, and only one pig was humanely euthanized at a particular time point from 6 hours to 28 days post-irradiation.
The safety of this 1064nm laser has been described in detail in another of my published articles, please refer to reference 25 (Fan, Y. et al. Quantitative and qualitative evaluation of recovery process of a 1064nm laser on laser-induced skin injury: in vivo experimental research. Laser Phys Lett 2019.). Literature research has revealed a scarcity of investigations concerning the effects of laser-induced skin damage and the attendant repair mechanisms. To delve deeper into the mechanisms governing laser-induced skin injury and subsequent repair, this study examined changes in collagen architecture and the expression of apoptosis-associated proteins, namely Bax, caspase-3, and caspase-9, within porcine skin wounds during the ensuing wound healing post laser irradiation.
The 1064nm laser has been widely used in clinical practice, and skin injuries caused by improper or excessive use as well as by high power irradiation are the focus of this study, and the relevant background has been revised in the manuscript. Some differences between laser irradiation and thermal burn in terms of skin damage and wound healing have been reported in the literature, which will be an important direction for conducting future studies.